# Physical Activity and the Quality of Life of Female Students of Universities in Poland

**DOI:** 10.3390/ijerph18105194

**Published:** 2021-05-13

**Authors:** Katarzyna Kotarska, Małgorzata Paczyńska-Jędrycka, Katarzyna Sygit, Kamila Kmieć, Aleksandra Czerw, Maria Alicja Nowak

**Affiliations:** 1Institute of Physical Culture Sciences, University of Szczecin, 71-065 Szczecin, Poland; katarzyna.kotarska@usz.edu.pl (K.K.); malgorzata.paczynska-jedrycka@usz.edu.pl (M.P.-J.); maria.nowak@usz.edu.pl (M.A.N.); 2Faculty of Health Sciences, Calisia University, 62-800 Kalisz, Poland; k.kmiec@akademia.kalisz.pl; 3Department of Health Economics and Medical Law, Faculty of Health Sciences, Medical University of Warsaw, 02-091 Warszawa, Poland; aleksandra.czerw@wum.edu.pl

**Keywords:** physical activity, quality of life, health training, women, WHOQOL-BREF questionnaire

## Abstract

Physical activity increases human health potential and has an impact on achieving a higher quality of life in society. The aim of our research was to determine the relationship between a physically active lifestyle and the quality of life of female students in the context of demographic and social factors (major, age, marital status, professional activity). The research was conducted among a group of 285 women studying physical culture and social sciences in Poznań and Szczecin (Poland). Average age: 22.7 ± 4.90. The standardized World Health Organization Quality of Life—BREF (WHQOL-BREF) questionnaire was used to assess the quality of life of female students, and the original survey technique was used to study the lifestyle of people undertaking physical activity in the context of socio-demographic factors. Nonparametric statistics were applied in the analyses of the results. The effect size was calculated for each test: E2R for the Kruskal–Wallis H test, Glass rank biserial correlation (rg) for the Mann–Whitney U test, and Cramér′s V for the χ2 test. The value of *p* ≤ 0.05 was assumed to be a significant difference. In the study, it was shown that a higher overall quality of life and health satisfaction, as well as better results in the physical, psychological, and environmental domains, were achieved by female students who assessed their lifestyle as physically active in comparison to those physically inactive. Higher scores of overall quality of life and satisfaction with health were found among female students of physical education and people participating in physical recreation, who also achieved better results in the environmental domain. Female students aged 23–25 had a higher quality of life in the physical, psychological, and social domains. Having a partner or spouse had a positive effect on the quality of life of female students defined by the social domain. A higher overall quality of life and satisfaction with health were characteristic of people who were employed. In the search of factors positively influencing the quality of life of society, it seems necessary to promote a physically active lifestyle among students. The observed differences in the quality of life and health satisfaction of female students of selected majors require targeted programs and interventions that improve the quality of their lives at various stages of their studies. Such activities increase the health potential of the individual and society, not only in the biological, but also psychosocial dimension.

## 1. Introduction

The World Health Organization (WHO, Geneva, Switzerland) defines quality of life as an individual′s perception of their position in life in the context of the culture and value systems in which they live and in relation to their goals, expectations, standards, and concerns (The World Health Organization Quality of Life–WHOQOL) [1]. Quality of life is perceived in the context of self-assessment on many levels (including physical, psychological, social, and spiritual), which is related to the individual level of health, satisfaction with life and health, and in relation to the physical, psychological, social, and environmental domains [2,3,4]. Quality of life has an effect on the wellbeing of an individual and his/her satisfaction with life [5]. The sense of quality of life and the factors influencing it evolve over the course of human life. Analyses conducted on the basis of the Social Diagnosis in Poland prove that the best predictors of the overall quality of life of Poles were the level of education, marriage, being an entrepreneur, or working in the public sector [6].

The quality of human life is related to the preferred lifestyle, which is also subject to constant changes. It includes everyday behavior, the characteristic ways of living of an individual or a social group. A feature that characterizes lifestyle is the possibility to make choices in certain spheres of life (e.g., education, work, family life, leisure time) above the breadline [7].

The health and physical activity of society are treated as a condition of basic professional and social competences, as well as the possibility of achieving a better quality of life. Physical activity is especially perceived as one of the most desirable forms of recreation and ways of spending leisure time due to its fundamental importance for health and the prevention of diseases of affluence. The use of leisure time by young people and representatives of various social groups, especially taking up its active forms (recreation, sport, tourism), is an important research subject. The diagnosis of real participation in these forms and the assessment of the general level of physical activity in society may constitute a measure of the consumption of leisure time as an essential element of modern human life [8].

Physical activity falls into the category of leisure time, which should be considered in the context of family and professional life, as well as in relation to the age of the respondents. Regular physical activity (recreational, sport) undertaken by women is a specific distinguishing featureof the lifestyle of an individual, as well as of a specific social group—a certain category of women. In the category of a physically active lifestyle, we observe the phenomenon of the interpenetration of a conscious choice of this behavior, leisure time, and life circumstances [9]. On the one hand, physical activity satisfies biological needs, and on the other, it supports social development [10]. It is also a tool for the health education of society; it shapes a physically active/sport lifestyle that is often associated with the choice of other pro-health behaviors (proper nutrition, avoiding smoking and consuming alcoholic beverages, systematic monitoring of one′s health) [11].

Lifestyle covers all behaviors of an individual or population, depending on his/her own interests, worldview, and hierarchy of values, as well as demographic factors [12]. A survey conducted at the beginning of 2019 on a representative random sample of 1858 Poles aged 15 and over showed that 64% of them declared physical activity at least once a month. This means that over one third of Poles did not undertake any physical activity, not even once a month. The most active were representatives of the young generation aged 15 to 24 (80%), persons in education (90%) or with higher education (78%), as well as Poles whose monthly income exceeded PLN 5000 (83%), above the national average [13].

Scientific research has repeatedly indicated the positive relationship between physical activity and people′s quality of life, health, and wellbeing [14]. The inhabitants of Wrocław aged 18–64 whose level of activity was assessed as high (at least 1500 METs/min/week) had a higher overall quality of life, as well as a higher perception of health and quality of life in the physical, psychological, social, and environmental domains. These relationships were weaker in the case of low physical activity. The chances of having a high overall perceived quality of life increase with growing levels of physical activity.

Quality of life and lifestyle (including physical activity) are determined by demographic, social, and economic factors. Some works emphasize the combined influence of many factors [15]. The research among Brazilian medical students indicates positive (male sex, self-efficacy, psychological and physical factors, including physical activity in leisure time, satisfaction with one′s body) and negative (female sex, economic status, place of residence, year of studies, chronic diseases, BMI, sleep problems, headaches) predictors of quality of life [16]. Similar results were also found among medical students in Pakistan and Chile [17,18].

The influence of peer groups, social norms, and environmental factors on the choice of specific behaviors is confirmed by research conducted in South Asia [19]. The level of physical activity, sleep intensity, and interpersonal, peer, and family relationships determined low scores of the quality of life of Chinese medical students [20]. High levels of anxiety and depression determined low scores of quality of life among Malay students of medical faculties [21]. Due to the unsatisfactory quality of life and health problems of students, it seems necessary to pay attention to their lifestyle and quality of life by applying a broad spectrum of research on health-promoting behaviors (including physical activity) of this social group. University students who are not related to physical culture and health show little interest in physical activity. There are concerns that groups of young, educated people not related to physical and health culture may avoid physical activity in the future and choose a physically inactive lifestyle that may be perceived as a role model, consequently lowering the quality of life of society. Universities of physical education, sport, tourism and recreation, and public health should intensify their efforts to ensure health for the individual and society. In the future, male and female students should promote a modern, physically active lifestyle that can contribute to improving the quality of life in society. In order to optimize our existence, it becomes necessary to monitor and diagnose conscious choices regarding physically active life in the context of quality of life, especially by women who undertake physical activity less frequently [17,18,22,23]. The aim of this work was to determine the relationship between a physically active lifestyle (assessment of lifestyle, physical activity, doing sports) and the quality of life of female students of selected majors in the context of demographic and social factors. A hypothesis regarding the relationship between a physically active lifestyle and the quality of life of female students was adopted. It was also assumed that major, age, marital status, and professional activity of the female respondents may modify this relationship.

## 2. Participants and Ethical Considerations

The research was conducted in the 2018/2019 academic year (October–December) among a group of 285 women studying physical education sciences (tourism and recreation–T&R, physical education–PE, public health–PH) and social sciences–SS in Poznań and Szczecin (Poland). Female first-cycle students accounted for 59.7%, and second-cycle students40.3%. According to research, 35.8% studied SS, 25.3% PH, 20.7% PE, and 18.2% T&R. The largest number of female students were aged 20–22 (48.1%), while the lowest number was aged ≥26 (11.6%). Most of the respondents were employed (62.5%).The female respondents most often lived in cities (68.1%).More than half (56.5%) had partners, and 7.7% were married. The female students declared having a physically active lifestyle and being rather physically active (75.4%). According to the study, 11.6% of the respondents practiced competitive sports at that time, and 68.4% of the women participated in physical recreation.

The inclusion criteria for the study group were as follows: age 18–35 years; female students of Polish universities; and fields of studies: physical education, public health, tourism and recreation, and social sciences. The criteria for exclusion from the study group were: age over 35, male gender, and women who were non-students or studying any other field of study than physical education, public health, tourism and recreation, and social sciences.

This study is a fragment of a largerstudy conducted as part of a research project on the health and quality of life of female students of universities in Poland. The first stage of the study (presented in the article) was carried out in two cities in Poland: Szczecin and Poznań. The authors of the study obtained the appropriate approvals to conduct research at the University of Szczecin and the Academy of Physical Education in Poznań, departments of physical education, public health, tourism and recreation, and social sciences (Bioethics Committee at the District Chamber of Physicians in Szczecin, Poland No. 15/KB/V/2015). All authors of the article were involved in collecting the data, while the data analysis was carried out by two authors of this paper.

The analyzed study programs envisage the implementation of classes in the field of physical activity, varying in terms of content and hours. First-cycle PE (physical education) students had 460 movement classes, while second-cycle students had 200 such classes. These included, among others, team sports games, gymnastics, athletics, and swimming. The T&R program included 200 h of classes during first-cycle studies and 60 h during second-cycle studies (including movement games, animation forms, activity in the aquatic environment, and health training). The PH study program included 60 h during first-cycle studies and 120 h during second-cycle studies (30 h are spent on physical education, and the rest for recreation in public health and dance classes). Female SS students completed 30 h of physical education during their first-cycle studies. There were no physical activities for them during second-cycle studies.

The research was approved by the Bioethics Committee at the District Chamber of Physicians in Szczecin No. 15/KB/V/2015.The research was conducted in accordance with the Helsinki Declaration of 1975. Written informed consent was obtained from each subject included in the study.

## 3. Data Collection

The standardized WHOQOL-BREF (World Health Organization Quality of Life—BREF) questionnaire was used to assess the quality of life of female students. The shortened version, adapted to Polish conditions in terms of language, culture, and psychometry, contained 26 questions [4]. The first two questions about the degree of life satisfaction and health were analyzed separately. The remaining questions covered four areas of quality of life: physical (pain and discomfort, drug and treatment dependence, energy and fatigue, mobility, rest and sleep, everyday tasks, ability to work);psychological (joy of life, meaning in life, concentration, body image, self-esteem, feeling of sadness);social (personal relationships, sexual activity, social support); and environmental elements (safety, neighborhood, financial resources, access to information, recreation and leisure time, housing, access to medical care, transport). Scoring for the areas was determined by taking the arithmetic mean for each of them. It reflects the individual perception of the quality of life in individual domains. The higher the number of points, the better the quality of life.

In order to demonstrate the relationship between a physically active lifestyle and quality of life in the context of socio-demographic variables, the diagnostic poll method was used. The original survey technique was applied to study the lifestyle of people undertaking physical activity (assessment of their lifestyle as being physically active, confirmed by participation in physical recreation and/or doing sports). The information was confirmed with an analysis of the study programs, especially in terms of planned and ongoing physical activities. For characteristics of the respondents and their assessment of their quality of life, socio-demographic variables (major, age, marital status, professional activity) were used.

## 4. Data Analysis

Before selecting the methods of statistical inference, the distribution normality of the variables was examined. In all cases, an abnormal distribution was found, and therefore, non-parametric statistics were used. Nonparametric statistics were applied in the analyses of the results. The Kruskal–Wallis test (H) was used to compare several independent samples. In the case of determining the statistical significance of differences for the comparison of two independent samples, the Mann–Whitney (U) test was employed. In qualitative analyses, the trait frequency and the independence chi-squared test were used.

The effect size was calculated for each test: E^2^_R_ for the Kruskal–Wallis H test, Glass rank biserial correlation (rg) for the Mann–Whitney U test, and Cramér′s V for the χ^2^ test. The value of *p* ≤0.05 was assumed to be significantly different. Statistical calculations were made with Statistica 13.1 for Windows (StatSoft Sp. zo.o., Krakow, Poland) and Microsoft Office Excel 2007 (Microsoft Sp. z o.o., Warsaw, Poland).

## 5. Results

According to the research, 285 women (mean age: 22.7 ± 4.90) participated in the study. Most female students were in the age group of 20–22 (48.1%). The greatest number of people aged ≤19 were female students of physical education (40.7%). The oldest (≥26 years) were female students of social sciences (24.5%) (*p* < 0.001 for the χ2 test, *p* = 0.3 for Cramér′s V). Female first-cycle students accounted for 59.7% (*p* < 0.001 for the χ2 test, *p* = 0.3 for Cramér′s V). Most of the respondents were employed (62.5%), lived in cities (68.1%), and achieved secondary education (66%) (*p* < 0.05 for the χ2 test, *p* = 0.2 for Cramér′s V). More than half of the surveyed women had partners without being in formal relationships (56.5%). Only 7.7% declared being married (*p* < 0.001 for the χ2 test, *p* = 0.2 for Cramér′s V) (Table 1).

The majority of female students declared having a physically active lifestyle and being rather physically active (75.4%) (Table 2). Female PE students most often assessed their lifestyle as physically active (66.1%), while SS students were the least likely to do so (14.7%). According to the study, 11.6% of the respondents practiced competitive sports at that time, most often studying PE (30.5%). Research analysis showed that 68.4% of women participated in physical recreation; most often, they werePE students (88.1%), and least often were SS students (52.0%). All female students who assessed their lifestyle as physically active or rather physically active undertook physical activity at a similar level (participation in physical recreation and doing sports—in total 80.0%). Female PE students more often participated in recreation and practiced competitive sports, as well as more often assessed their lifestyle as physically active (*p* < 0.001 for the χ2 test), but the effect size for these relationships was average (*p* = 0.3 for Cramér′s V).

General diversity was found between satisfaction with life and health and the assessment of one′s lifestyle (*p* < 0.001, E^2^_R_ = 0.06; *p* < 0.001, E^2^_R_ = 0.07, respectively) (Table 3). People who evaluated their lifestyle as physically active or rather physically active assessed their satisfaction with life and satisfaction with health more highly than those evaluating theirlifestyle as inactive (*p* < 0.001; *p* < 0.01 for the U test; *p* < 0.001; *p* < 0.05 for the U test, respectively). These differences were confirmed by the above-average effect strength (rg = 0.4). Moreover, female students who evaluated their lifestyle as physically active assessed their satisfaction with life and health more highly compared to those who evaluated their lifestyle as rather physically active (effect strength below average). Overall diversity was observed in the physical, psychological, and environmental domains. In the physical domain, higher scores were achieved by female students who assessed their lifestyle as rather physically active compared to perceiving it as inactive (*p* < 0.05 for the U test). In the psychological domain, female students who perceived their lifestyle as physically or rather physically active achieved higher scores compared to inactive ones (*p* < 0.01 for the U test U, rg = 0.3; *p* < 0.01 for the U test, respectively). In the environmental domain, the lowest results were achieved by female students who perceived their lifestyle as inactivecompared to physically active and rather active (*p* < 0.01; *p* < 0.01 for the U test, respectively). There was no such diversity in the social domain.

The female respondents participating in physical recreation had a higher assessment of quality of life and health, and better results in the environmental domain compared to physically passive women (*p* < 0.01; *p* < 0.05; *p* < 0.05 for the U test, respectively). Similarly, women who declared doing sports had higher satisfaction with their health compared to those not doing sports (*p* < 0.05 for the U test), but the effect strength was small (rg = 0.1–0.2) (Table 4).

There was no general difference in the assessment of overall quality of life between female students of selected majors. These differences were observed between the assessment of quality of life by female PE and SS students (*p* < 0.01 for the U test; rg = 0.4) (Table 5). Female PE students rated their quality of life higher (the strength of the effect was above average). A general differentiation was found in the assessment of satisfaction with health by the respondents (*p* < 0.05 for the H test). Female PE students were also more satisfied with health than female SS students (*p* < 0.01 for the U test). This was confirmed by the high strength of the effect (rg = 0.5). The general differentiation between women of selected majors in the physical domain was confirmed (*p* < 0.05 for the H test). Differences were observed between women studying PH and PE and PH and SS (*p* < 0.05, *p* < 0.05 for the U test; rg = 0.4, respectively). Female PH students, compared to female PE and SS students, achieved higher results in the physical domain (above average effect strength). Pairwise comparisons indicated differences in the physical domain between female T&R and SS students (*p*< 0.05 for the U test; rg = 0.4). Female T&R students achieved higher scores in the environmental domain. There were no differences in the psychological and social domains.

The overall quality of life and health satisfaction of the female students were not related to their age; however, the general differentiation in the physical, psychological, and social domains was confirmed (*p*<0.001; *p*<0.01; *p*<0.05 for the H-test, respectively) (Table 6). Women aged 23–25 years had better results in the physical domain compared to those aged ≤19, 20–22, and ≥26 (*p*<0.001, *p*<0.01, *p*<0.05 for the U test, respectively). In the psychological domain, women aged 23–25 had higher scores compared to the group of ≤19 year olds (*p*<0.001 for the U test; rg =−0.4) and 20–22 year olds (*p*<0.01 for the U test; rg = −0.2). Female students aged 23–25 achieved higher results compared to female respondents aged ≤19 in the social domain (*p*<0.01 for the U test; rg = −0.3).

There was a general differentiation between the social domain of quality of life of female students by marital status (*p*<0.001 for the H test; E^2^_R=_0.2). Unmarried female students achieved lower results compared to married women and women with partners (*p*<0.001 for the U test, rg = −0.5; *p*<0.01 for the U test, rg =−0.4) (Table 7). These differences were confirmed by the effect strength.

Female students undertaking work had higher scores of overall quality of life and health satisfaction compared to those non-working (*p*<0.05; *p*<0.001 for the U test, respectively), however, the effect size was small (Table 8). In terms of individual areas of quality of life, these differences were not found.

## 6. Discussion

When undertaking the study, ahypothesis regarding the relationship between a physically active lifestyle and the quality of life of female students was adopted. It was also assumed that major, age, marital status, and professional activity of the female respondents may modify this relationship. Relationships between the assessment of one′s lifestyle and the quality of life of female students of selected majors (physical education, tourism and recreation, public health, social sciences) was observed. The assessment by female students of their lifestyle as physically active or rather physically active was correlated with a better overall quality of life in particular domains (except for the social domain) and greater satisfaction with health. There were no differences in terms of personal relationships, sexual activity, and social support (social domain). The majority of female students assessing their lifestyle as physically active participated in physical recreation, and over 1/10 of them practiced competitive sports. Earlier studies of physical activity of female students of tourism and recreation and physical education confirmed greater physical activity and more frequent practicing of sports by female students of physical education [24].

In nationwide diagnostic studies, it was found that people who undertook any kind of physical activity achieved higher health scores [6]. It was also noticed that students who regularly practiced aerobic exercise and strength training had a higher quality of life than those physically inactive. It was also demonstrated that a significant modifier of quality of life is high self-esteem resulting from physical activity [25,26,27]. Studies among female students of various majors (including social sciences) indicated that a healthy lifestyle (including activity and physical fitness) had an effect on a higher evaluation of quality of life [28]. Students who were engaged in physical activity more typical for their lifestyle experienced greater satisfaction with life [29]. A high level of quality of life was observed among students of various majors from Lviv who had a higher level of physical activity [30]. Studies among Croatian youth show a higher level of physical activity among physiotherapy students and their higher quality of life (in the physical and social fields) compared to social sciences students who had lower levels of physical activity, but better results in terms of the mentality domain [26]. At the same time, there are studies regarding students of physiotherapy that do not confirm the relationship between a very high level of physical activity of respondents and their self-esteem and quality of life, regardless of gender [31]. However, a significant positive correlation between high levels of physical activity and quality of life (in all domains) was found among Brazilian medical students (both sexes). For students with low levels of physical activity, the correlation was only significant in the case of the physical and social domains [32].

Differences in the quality of life of female students of selected majors were observed. Physical education female students had a higher overall evaluation of quality of life and expressed greater satisfaction with health compared to those studying social sciences. Public health female students achieved higher results in the physical domain than those studying physical education and social sciences. In the environmental domain, female T&R students scored higher than female students in social sciences. Female social science students had the lowest evaluation for overall quality of life, as well as in the physical and environmental domains, and were the least satisfied with their health. Higher results of public health female students in the physical domain (pain and discomfort, drug and treatment dependency, energy and fatigue, mobility, rest and sleep, everyday activities, ability to work) may result from better coping with various situations by these female students, who, as a result of their preparation, are more aware of how their bodies work and are more prone to pro-health behavioral choices. Similarly, higher scores of female T&R students in the environmental domain (safety, neighborhood, financial resources, access to information, recreation and leisure time, housing, access to medical care, transport) are probably the result of studies in this field.

Comparisons of students of physical education and other faculties of the T.C.Hitit University (Corum, Turkey) showed that quality of life, along with the physical, mental, social, and environmental domains, was higher among people studying physical education [33]. It has also been shown that the probability of a high level of perceived overall quality of life rises with the increase in the level of physical activity of people aged 18–64 [14]. Physical activity and sleep duration had a positive effect on the quality of life of Chinese medical students [34]. Studies of Polish students of various majors (public health, physiotherapy, tourism and recreation, psychology, pedagogy, and theology) indicated that only some types of physical activity show a positive relationship with quality of life. Physical activity in the household was most positively and significantly correlated with quality of life [35]. It should be noted that women are more active in housework, and men are more active in their leisure time [36]. Housework comprises activities that result from daily duties. Despite the physical exertion associated with them, this activity cannot be classified as a lifestyle, the essence of which is having a choice. Despite some inconsistencies of the results of various studies resulting from the comparison of female students of many majorsin different periods of time, often using different methods, there is a relationship between the choices regarding a physically active lifestyle and quality of life.

It was confirmed that socio-demographic factors (age, marital status, and professional activity) modify the relationship between the choice of a physically active lifestyle and quality of life of female students of selected majors. In the study, it was observed that women aged 23–25 assessed their quality of life the highest in the physical, psychological, and social domains, while the youngest students aged ≤19 assessed them the lowest. Additionally, a tendency towards lower quality of life scores among women aged ≤26 was noticed. These results have been partially confirmed by other studies [5]. Senior students perceived quality of life in the physical and environmental domains as lower. The results in the physical domain were significantly higher for the fourth year respondents compared to the first year (younger) ones. International studies of nursing students (Chile, Egypt, Greece, Hong Kong, India, Kenya, Oman, Saudi Arabia, and the USA) showed that they achieved the highest level of quality of life in the physical domain and the lowest in the social domain. The main determinants of quality of life include, among others, age [37].

It was observed that the quality of life of female students in the social domain increased with being in a formal or informal relationship. A study of dentistry students in the United States showed that single students (especially men) perceived quality of life lower than married students [5].The quality of life of medical students improved among married people [38]. Research conducted among Chinese medical students (without a gender breakdown) shows that, among others, family satisfaction had an effect on quality of life [20].

It was noticed that female students undertaking paid work were characterized by a higher quality of life and health satisfaction. However, there were no differences between working and non-working women in the particular domains of quality of life. Research has shown that increasing working hours has a negative impact on academic performance. Students who work full-time while studying are less likely to complete their studies than those who work part-time ordo not work at all [39].

The obtained research results show the importance of physical activity for the quality of life of the respondents. The study confirmed other studies conducted by researchers in various countries around the world, as outlined above. However, a comparison of the obtained results with the research conducted in the United States, among others, deserves attention. The results of a study conducted at US universities showed that women taking up sports (regardless of the chosen field of study) displayed higher health-related satisfaction. Particular attention was paid to the consistency of undertaken physical exercises and the choice of various forms of physical activity [40]. Research conducted in France also proved the impact of various forms of physical activity on overall quality of life, as well as on health satisfaction [41]. The impact of physical activity on health is a widely studied topic among Chinese students. Our study confirmed that socio-demographic factors have a significant impact on quality of life. The research carried out by Chinese researchers showed (and confirmed the results of this study) that female students who were not in a relationship and did not take up paid work were characterized by a lower quality of life [42].

In search of factors positively influencing the quality of life of society, it seems necessary to promote a physically active lifestyle (regular physical activity, participation in physical recreation, doing sports) not only among students of physical culture sciences and public health, but also other majors. The observed differences in the quality of life and satisfaction with health of female students of particular majors require targeted programs improving the quality of life of female students at various stages of their studies. Such activities increase the health potential of the individual and society, not only in the biological, but also psychosocial dimension.

The use of standardized questionnaires in further research will allow for a wider observation of the quality of life of women. In addition, increasing the number of respondents would help in further determining the quality of life of the respondents, as well as in assessing the impact of an active lifestyle on quality of life, which the respondents considered crucial for strengthening and improving health. After extensive research in the group of university students, more constructive conclusions can be drawn that would allow specific preventive measures to be taken from an early age in order to change incorrect health behaviors (especially increasing physical activity), and thus improve the health and quality of women’s lives.

## 7. Conclusions

A higher overall quality of life and satisfaction with health of female students evaluating their lifestyle as physically active compared to those who were inactive was indicated in the research. Better results in the physical, psychological, and environmental domains of women with a physically active lifestyle were also confirmed. Higher scores for overall quality of life and in the environmental domain, as well as greater satisfaction with health, were characteristic of female students participating in physical recreation.

The impact of the selected descriptive variables on overall quality of life was observed. Higher scores for overall quality of life and health satisfaction were seen among female PE students. PH students scored better in the physical sphere, while T&R students were better in the environmental sphere. Students aged 23–25 performed better in physical, psychological, and social domains. The influence of respondents’ marital status on quality of life was also observed: having a partner or a spouse had a positive effect on the quality of life of female students, defined by the social domain. Students who took up gainful employment were characterized by a high overall quality of life and health satisfaction.

## Figures and Tables

**Table 1 ijerph-18-05194-t001:** Socio-demographic characteristics of students (independence χ^2^ test and Cramér′s V.

Variables	Major				Total (285)		*p* for χ^2^	Cramér’s V
	T&R (*n* = 52)	PE (*n* = 59)	PH (*n* = 72)	SS (*n* = 102)	*n*	%		
**Age**							*p* < 0.001	0.3
≤19	11.6	40.7	15.3	1.0	42	14.7		
20–22	44.2	44.1	52.8	49.0	137	48.1		
23–25	42.3	13.6	22.2	25.5	73	25.6		
≥26	1.9	1.7	9.7	24.5	33	11.6		
**Degree of Study**							*p* < 0.001	0.3
First-cycle	55.8	84.8	65.3	43.1	170	59.7		
Second-cycle	44.2	15.2	34.7	56.9	115	40.3		
**Gainful Employment**							n.s.	-
Yes	61.5	64.4	54.2	67.7	178	62.5		
No	38.5	35.6	45.8	32.2	107	37.5		
**Place of Residence**							n.s.	-
City <100 thousand	17.7	34.5	20.8	28.7	73	25.9		
City >100 thousand	58.8	41.4	45.8	31.7	119	42.2		
Village	23.5	24.1	33.4	39.6	90	31.9		
**Marital Status**							*p* < 0.001	0.2
I am not in a relationship	50.0	39.0	41.7	22.6	102	35.8		
I have a partner	48.1	55.9	55.7	61.8	161	56.5		
Married	1.9	5.1	2.8	15.7	22	7.7		

* n.s., not significant; T&R, tourism and recreation; PE, physical education; PH, public health; SS, social sciences.

**Table 2 ijerph-18-05194-t002:** Physically active lifestyle as assessed by the female respondents, participation in physical recreation, and doing sports by major (independence χ^2^ test and Cramér′s V).

Variables	Major				Total (285)		*p* for χ^2^	Cramér’s V
	T&R * (*n* =52)	PE * (*n* =59)	PH * (*n* =72)	SS * (*n* =102)	*n*	%		
**Assessment of One’s Lifestyle**							*p* < 0.001	0.3
As physical active	25.0	66.1	20.8	14.7	82	28.8		
As rather physically active	53.9	30.5	52.8	49.0	134	47.0		
As not physically active	21.1	3.4	26.4	36.3	69	24.2		
**Competitive Sport**							*p* < 0.001	0.3
Yes	9.6	30.5	4.2	6.9	33	11.6		
No	90.4	69.5	95.8	93.1	252	88.4		
**Participation in Physical Recreation**							*p* < 0.001	0.3
Yes	80.8	88.1	66.7	52.0	195	68.4		
No	19.2	11.9	33.3	48.0	90	31.6		

* T&R, tourism and recreation; PE, physical education; PH, public health; SS, social sciences.

**Table 3 ijerph-18-05194-t003:** Satisfaction with life, health, and particular domains of quality of life (WHOQOL-REF) of female students by a physically active or not physically active lifestyle (U test, rg).

Specification	Assessment of One’s Lifestyle	Value of *p* for U Statistics		Glass Rank Biserial Correlation (rg)		Rank Means
		As Rather Physically Active	As Not Physically Active	As Rather Physically Active	As Not Physically Active	
Satisfaction with LifeH(2279) = 17.15E^2^_R_ = 0.06 *p* < 0.001	As Physically Active	0.034	0.001	0.2	0.4	161.94
	As Rather Physically Active		0.006		0.2	140.93
	As Not Physically Active					111.96
Satisfaction with HealthH(2279) = 20.81 E^2^_R_ = 0.07 *p* < 0.001	As Physically Active	0.002	0.000	0.2	0.4	167.30
	As Rather Physically Active		0.022		0.2	137.28
	As not Physically Active					112.76
Physical DomainH(2285) = 5.75 E^2^_R_ = 0.02 *p* < 0.05	As Physically Active	0.714	0.059	−0.1	0.2	147.13
	As Rather Physically Active		0.021		0.2	150.99
	As Not Physically Active					122.57
Psychological Domain H(2285) = 11.0 E^2^_R_ = 0.03 *p* < 0.001	As Physically Active	0.397	0.002	0.1	0.3	157.38
	As Rather Physically Active		0.004		0.2	148.49
	As Not Physically Active					115.25
Social Domain H(2285) = 0.34 E^2^_R_ = 0.01 n.s.	As Physically Active	0.567	0.874	−0.1	−0.1	139.40
	As Rather Physically Active		0.733		0.1	145.89
	As Not Physically Active					141.67
Environmental DomainH(2285) = 9.29 E^2^_R_ = 0.01 *p* < 0.05	As Physically Active	0.869	0.010	0.1	0.2	152.21
	As Rather Physically Active		0.004		0.2	150.86
	As Not Physically Active					116.78

**Table 4 ijerph-18-05194-t004:** Satisfaction with life, health, and particular domains of quality of life (WHOQOL-BREF) of female students by participation in physical recreation and sports (U test, rg).

Specification		Answers	Value of *p* for U Statistics	Glass Rank Biserial Correlation (rg)	Rank Means
**Physical Activity**	Satisfaction with life	Yes	0.010	0.2	147.44
		No			123.83
	Satisfaction with health	Yes	0.028	0.1	146.49
		No			125.90
	Environmental domain	Yes	0.025	0.2	150.36
		No			127.03
**Doing Competitive Sport**	Satisfaction with health	Yes	0.016	0.2	168.96

**Table 5 ijerph-18-05194-t005:** Satisfaction with life, health, and particular domains of quality of life (WHOQOL-BREF) of female students by major (H test, E^2^_R_, U test, rg).

Specification	Majors	PH *	PE *	SS *	PH *	PE *	SS *	Rank Means
		Value of *p* for U Statistics			Glass Rank Biserial Correlation (rg)			
Satisfaction with Life H(3279) = 6.65 E^2^_R_ = 0.02 n.s.	T&R	0.890	0.134	0.356	0.1	−0.4	0.2	140.37
	PH		0.086	0.399		−0.3	0.1	138.57
	PE			0.011			0.4	160.29
	SS							129.04
Satisfaction with Health H(3, 279) = 9.61 E^2^_R_ = 0.03 *p* < 0.05	T&R	0.860	0.228	0.107	0.1	−0.3	0.3	144.42
	PH		0.141	0.114		−0.3	0.2	142.12
	PE			0.002			0.5	160.75
	SS							124.2
Physical Domain H(3285) = 8.61 E^2^_R_ = 0.03 *p* < 0.05	T&R	0.657	0.105	0.067	−0.1	0.4	0.3	155.74
	PH		0.035	0.014		0.4	0.4	161.52
	PE			0.780			0.1	132.05
	SS							129.75
Environmental Domain H(3, 285) = 6.45 E^2^_R_ = 0.02 n.s.	T&R	0.580	0.216	0.016	0.2	0.3	0.4	161.10
	PH		0.539	0.092		0.1	0.3	150.95
	PE			0.258			0.2	142.67
	SS							128.33

* T&R, tourism and recreation; PE, physical education; PH, public health; SS, social sciences.

**Table 6 ijerph-18-05194-t006:** Satisfaction with life, health, and particular domains of quality of life (WHOQOL-BREF) of female students by age (H test, E^2^
_R_, U test, rg).

Specification	Age	20–22	23–25	≥26	20–22	23–25	≥26	Rank Means
		Value of *p* for U Statistics			Glass Rank Biserial Correlation (rg)			
Physical Domain H(3285) = 16.3432 E^2^_R_ = 0.06 *p* < 0.001	≤19	0.068	0.000	0.499	−0.2	−0.4	−0.1	112.58
	20–22		0.008	0.416		−0.2	0.1	140.39
	23–25			0.013			0.3	172.30
	≥26							127.69
Psychological Domain H(3, 285) = 13.91 E^2^_R_ = 0.05 *p* < 0.001	≤19	0.108	0.000	0.154	−0.2	−0.4	−0.2	113.98
	20–22		0.005	0.789		−0.2	−0.1	137.54
	23–25			0.119			0.2	170.28
	≥26							142.22
Social Domain H(3, 285) = 7.95 E^2^_R_ = 0.03 *p* < 0.05	≤19	0.161	0.009	0.605	−0.1	−0.3	−0.1	121.59
	20–22		0.080	0.496		−0.2	0.1	141.93
	23–25			0.053			0.2	162.85
	≥26							130.75

**Table 7 ijerph-18-05194-t007:** Satisfaction with life, health, and particular domains of quality of life (WHOQOL-BREF) of female students by marital status (H test, E^2^_R_, U test, rg).

Specification	Marital Status	Married Woman	Having a Partner	Married Woman	Having a Partner	Rank Means
		Value of *p* for U Statistics		Glass Rank Biserial Correlation (rg)		
Social Domain H(3, 285) = 49.31E^2^_R_=0.2 *p*<0.001	Single	0.000	0.005	−0.5	−0.4	97.82
	Married Woman		0.430		0.1	170.13
	Having a Partner					153.88

**Table 8 ijerph-18-05194-t008:** Satisfaction with life, health, and particular domains of quality of life (WHOQOL-BREF) of female students by gainful employment/professional activity (H test, E^2^_R_, U test, rg).

Specification	Gainful Employment	No	No	Rank Means
		Value of *p* for U Statistics	Glass Rank Biserial Correlation (rg)	
Satisfaction with Life	Yes	0.025	0.1	147.71
	No			127.59
Satisfaction with Health	Yes	0.000	0.2	151.72
	No			121.15

## Data Availability

Data are not publicly available and data sharing is not applicable to this article.

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
