# Peer review of "Physical Activity and the Quality of Life of Female Students of Universities in Poland"

_ijerph, 2021, doi:10.3390/ijerph18105194_

Round 1

Reviewer 1 Report

Dear authors:

The article is interesting and explores a current and pertinent phenomenon.
The introduction provides an understanding of what is known about the topic and justifies the need for this study. The objective is clear.
The method needs to be clarified:
1 - Criteria for the inclusion of participants;
2 - How the sample size was calculated;
3 - Which methodological procedures were used in data collection - how participants accessed the instrument, how many researchers were involved in data collection, how many were involved in data analysis;
4 - How was the test used to evaluate the normality distribution of the population - to support the option for non-parametric tests;
5 - Present in the method some percentages regarding the characterization of the sample that belong to the results;

Regarding the results:
Present the descriptive statistics related to the sociodemographic data of the population and the results of the application of the instruments.

It is not clear how the Cramer's V test is used in the 1st table.

In the discussion it is possible to improve the dialogue with the results of other studies to understand why these results in this Polish population.

The final list of references should be revised, standard abbreviations of the journals are not always used, there are data of the publications missing in some references (number and volume of the journal), among others.

Author Response

Dear Reviewer,

the letter is attached.

Reviewer 2 Report

The manuscript entitled "Physical activity and the quality of life of female students of universities in Poland" is generally well-written and presents relevant results considering the Poland university students. I suggest some minor corrections before the manuscript accepted for publication.

Specific comments

. Abstract: "... ± 4.90 years."

. Introduction, Line 103-104: Have you tested men students? If not, withdraw this last sentence.

. Introduction, Line 125 and 128: These two small paragraphs are not necessary. Rewrite in the previous.

. Methodology: considering space between numbers and units or numbers and symbols along with the manuscript.

. Results: The authors should standardize the use of abbreviations.

. Results: The use of "statistically significant" is not needed. Consider this change in the overall manuscript. 

. Results: Check tables first and sometimes second column length. It is not well presented.

. Discussion: Consider adding the hypothesis formulation and results test.

. Conclusions and Limitations: I suggest that the study limitations should be written in the previous section. I also suggest the author be brief in the Conclusions presentation.

Author Response

Dear Reviewer,

the letter is attached.

Round 2

Reviewer 1 Report

Dear authors:

I congratulate you for the article and the work developed.

Best regards;